# Exploring a Climate Gradient of Midwestern USA Agricultural Landscape Runoff Using Deuterium (δD) and Oxygen-18 (δ$^{18}$O)

**Lu Zhang** [1,2]**, Joe Magner** [2,]*** and Jeffrey Strock** [3]

1 Stantec Engineering, Austin, TX 78723, USA; lu.zhang@stantec.com
2 Department of Bioproducts and Biosystems Engineering, University of Minnesota, St. Paul, MN 55108, USA
3 Department of Soil Water and Climate, Southwest Research and Outreach Center, University of Minnesota, Lamberton, MN 56152, USA; jstrock@umn.edu
* Correspondence: jmagner@umn.edu

**Abstract:** Intensive agricultural activities have altered the hydrology of many Midwestern USA landscapes. Drain tiles (subsurface corrugated and perforated flexible tubing) has changed how and when water is discharged into the streams. Stable isotopes of oxygen (oxygen-18) and hydrogen (deuterium) were used to investigate hydrologic characteristics of three intensively managed agricultural landscapes in southern Minnesota (MN) and South Dakota (SD). Monthly δD and δ$^{18}$O samples were collected across a climatic gradient from March 2016 to March 2018. Local meteoric water lines were established for the comparison of precipitation and evaporation magnitude from different sources at each location. These included vadose zone, phreatic zone, deep groundwater, tile drain, and river source waters. Two end-member hydrograph separation was performed at each site on selected dates to partition the shallow groundwater tile drainage contribution to streamflow. A lumped parameter modeling approach was applied to each dataset to investigate the mean transit time of water through different zones. Local meteoric water lines demonstrated differences in isotopic signatures due to the climate gradient to show the impact of low humidity and less rainfall. The hydrograph separation results showed that, from west South Dakota to eastern Minnesota, tile drains contributed about 49%, 64%, and 50% of the watershed streamflow. Precipitation took an average of 9 months to move through different pathways to end up in groundwater and an average of 4 months to end up in tile drains. This study confirms the important role tile drains play in the intensively managed fields and watersheds of Midwestern agriculture.

**Keywords:** isotopes; tile drains; hydrological pathways

---

## 1. Introduction

Extensive tile drain system implementation altered the local surface hydrology by allowing rapid subsurface water discharge into the receiving waters [1]. Schottler and others [2] examined 21 watersheds across the Minnesota River basin and concluded that watersheds with large land use changes experienced increased in season and annual water yields—greater than 50% since 1940. This increase is highly correlated with drainage and the concordant loss of depressional areas for water storage.

The major negative impacts of tile drains on field and small watershed hydrology are less residence time in vadose zone storage and flashier stream hydrographs. Indirectly, tile drains could also reduce evapotranspiration (ET) and reduce water availability for plants, thus reducing crop production. Increased stream peak flow can cause negative downstream impacts, such as flooding and increased streambank erosion. Less water holding time in the soil profile also allows more nutrients into the surface water with less hydraulic residence time (HRT), which limits the biological nutrient removal processes in the soil. Increasing soil water holding is important for the three sites as the Cottonwood River was listed on 303(d) as impaired for turbidity (TSS) and Le Sueur River was listed as impaired for excess

nutrients. Vermillion River was impaired for TSS as indicated by the Vermillion River Strategic Plan [3].

The pathways and mechanisms responsible for this altered hydrology are poorly quantified across these watersheds at varying scales. Hydrograph separation is one of the methods widely used to study the source of streamflow and the contribution from each source. Hydrograph separation helps quantify the percent contribution of each end member. Two-component hydrograph separation is the most commonly used. It means two water sources are accounted for to explain the sources of water in a stream. When more constituents are measured, more components/sources can be included to better describe watershed processes.

Hydrograph separation method has been used in agricultural settings. It is commonly performed with environmental tracers, conservative tracers, and semi-conservative and reactive tracers [4]. Conservative tracers are tracers that do not get affected by biogeochemical processes during transport, such as chloride and oxygen stable isotope. Non-conservative tracers, such as nitrate, changes concentration during the transport. Numerous studies have investigated watershed characteristics using tracers and hydrograph separation.

Rozemeijer and others [5] calculated that the tile drain contribution to the total ditch discharge decreased from 80% to 28% in response to a rainfall event. Tomer and others [6] concluded that 68.6% of annual outflow came from tile drains, compared to a SWAT prediction of 71%, to confirm the validity of the method.

Tomer [6] used the hydrograph separation method to separate tile-outflow hydrographs into discharge from tile drain and surface intake component for water quality purposes at Tipton Creek watershed, Iowa. They concluded that surface intakes were responsible for 13% of tile drain discharge and subsurface tile drains dominated the nitrate delivery based on the nitrate concentration measurement at selected locations.

Schilling and Helmers [7] examined the hydrographs of tile-drained watershed in the Walnut Creek watershed in Iowa using hydrologic models. They concluded that tile drainage primarily affects the baseflow portion of a hydrograph and tile drainage increased the groundwater contribution to surface water.

Contribution of tile drainage to stream varies based on location. Amado [8] found that tile drain contributes approximately 15% to 43% of total annual streamflow in northeast Iowa using end-member analysis with nitrate concentrations. This value is dependent on precipitation. In a wet year, excessive precipitation allows more volume of water to infiltration, increasing discharge from tiles. This can cause a higher percent contribution of tile flow to streamflow.

Hydrogen and oxygen stable isotopes tracers have been used in water science related research to help understand hydrologic pathway and mixing mechanisms. They have also been used to study lake evaporation, waterbody residence time, and extreme event impacts on local hydrology [9–13].

Deuterium (D) is one of the two stable isotopes of hydrogen. It is heavier than protium. Hydrogen also has a third isotope, tritium, which is the heaviest and radioactive. The half-life of tritium is 12.3 years. Protium comprises about 99.985% of the hydrogen atoms in the atmosphere, whereas deuterium only accounts for 0.015% [14]. Oxygen has three isotopes as well: $^{16}O$ (99.895%), $^{17}O$ (0.038%), and $^{18}O$ (0.2%). All three isotopes are stable.

Vienna Standard Mean Ocean Water (VSMOW) is a universal standard set by International Atomic Energy Agency (IAEA) for comparison of heavy isotopes as they are usually in trace amount. The following equations are used for calculating the relative abundance of D and $^{18}O$:

$$\delta D \ (\text{‰}) = \ (\frac{R_{sample}}{R_{VSMOW}} - 1) * 1000 \tag{1}$$

$$\delta^{18}O \ (\text{‰}) = \ (\frac{R_{sample}}{R_{VSMOW}} - 1) * 1000 \tag{2}$$

where $R$ equals to $^{2}_{1}H/^{1}_{1}H$ and $^{18}_{8}O/^{16}_{8}O$, respectively, and $\delta$ represents the ratio.

Meteoric water lines (MWL) were established by plotting $\delta D$ against $\delta^{18}O$ from meteoric water [15], also known as precipitation; the global MWL shows data from across the globe with an average slope of 8 [14]. When plotting samples from one region, the local MWL tends to deviate from the global average due to the source of the water vapor. The ocean is enriched in $^{18}O$ as $^{16}O$ evaporates at a slightly faster rate. When tropical clouds move north, precipitation along the way removes slightly more $^{18}O$ from the clouds than $^{16}O$. Polar clouds at a higher latitude tend to be lighter or more depleted in $^{18}O$. Therefore, the polar region waters are richer in $^{16}O$. If the precipitation comes from clouds near the equator, the moisture tends to be heavier or more enriched in $^{18}O$. The points on the local MWL are scattered along the line depending on the mix of gulf or polar air masses. The slope of the line is due to further fractionation during precipitation. Local MWL for all three sites were established with the available precipitation data.

When plotting surface water isotopic signatures on the MWL, some points will deviate from the main regression line to the right. which represents residual water after evaporation. The further along this line the data plots, the more evaporation that likely occurred.

Stable oxygen and hydrogen isotopes have also been used to help understand the HRT/mean transit time (MTT) of water [16–18]. The two terms are usually not very well distinguished in the literature, and they have been used interchangeably [19]. The two terms represent different processes in a catchment system. Hydraulic residence time is defined as the length of time a molecule spends in a catchment system from entry [20]. Transit time is the time that it takes a molecule to exit the catchment system [21–23]. For example, in soil water storage, water molecules that are retained in the soil have a longer residence time than water that drained out. This is accounted for conceptually in the mean HRT, but not MTT, as it does not exit the system. However, these two terms are also highly related. The transit time of water through a system highly depends on the hydraulic pathway within the system itself. A longer transit time may indicate longer residence time. Therefore, investigating MTT will give an indication of HRT and, thus, reflect on the water quality benefit of the system.

Hrachowitz [24] used chloride as the conservative tracer to study transit times of two small (~1 km$^2$) watersheds using a lumped parameter approach. Weekly chloride concentration in precipitation and streamflow was collected for 8 years and corrected for evaporation. The authors used exponential flow, exponential piston flow, diffusion/dispersion, gamma models, as well as sine wave model. They concluded that the shorter MTT helps to fully recover the tracers and increases confidence in the feasibility of the travel time distributions and the validity of the assumptions [24,25].

Common simplification of this lumped parameter model takes advantage of the seasonal variation of isotopic composition and can be represented with a sine-wave model. However, this model does not allow variation of different flow types [19].

Burns and McDonnell [12], in their paper, used the sine-wave model to estimate stream water, groundwater, and soil water residence times based on annual isotope signature from surface water and meteoric water to investigate impacts of a beaver pond on runoff processes. They measured $^{18}O$ monthly from June 1989 to December 1990. The mean residence time was used. The residence time in this model was defined as the average time that "elapses between parcels of water entering as precipitation and leaving again as streamflow" [16], which is the same as MTT. A longer MTT indicates greater damping of seasonal tracer cycles. Therefore, the amplitude of a seasonal tracer is used for such a model [16]. The model produced reasonable results and helped authors conclude that a beaver pond had no significant influence on baseflow.

The tile drain contribution to streamflow has not been well documented for the South Dakota and Minnesota regions. The question this study seeks to answer is how the tile drain system affects local field scale hydrology, and what effect climate gradients have on streamflow sources. Stable isotope tracers were used in this study to perform event hydrograph separation and mean transit time (MTT) estimation at each site. Major end

members were assumed to be tile drain, deep groundwater, and/or vadose zone water. A sensitivity analysis was completed for each site to test for the impact of each parameter.

## 2. Study Area

The Cottonwood River and Le Sueur River Watersheds in Minnesota (MN) and Vermillion River Watershed in South Dakota (SD) are intensively managed agricultural watersheds (Figure 1). Cropland accounts for 88% of the land use within the Cottonwood River Watershed [26,27], 82% for Le Sueur River Watershed [28], and 67% for Vermillion River Watershed [29].

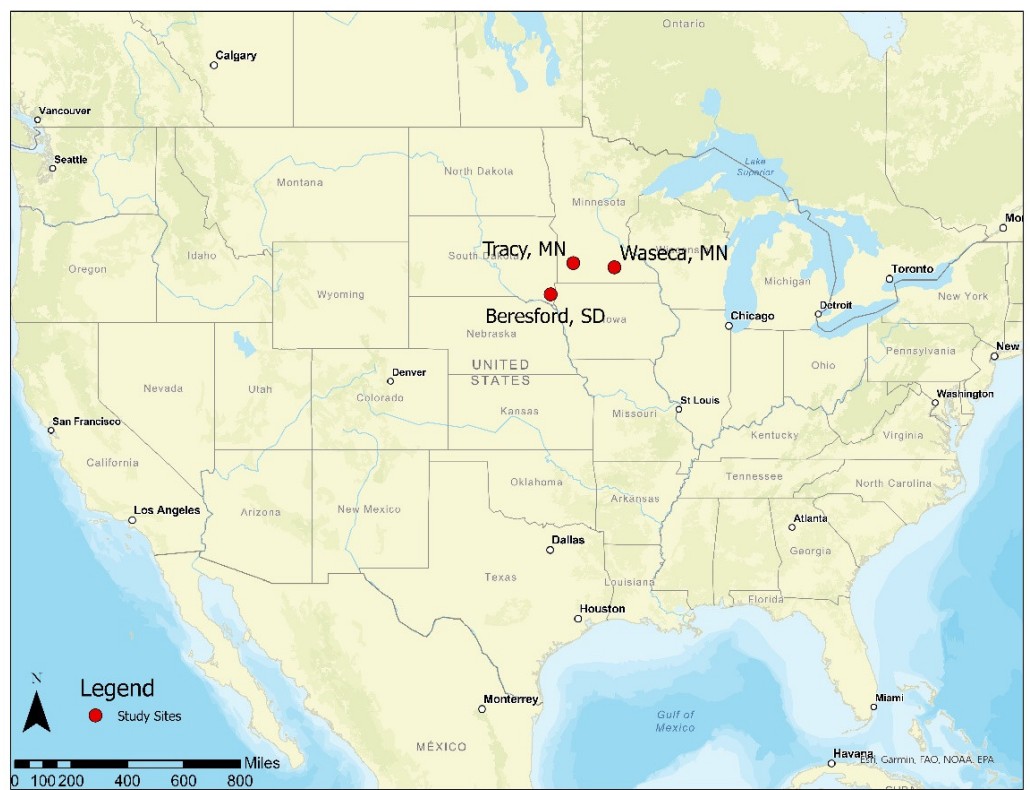

**Figure 1.** Study sites: Beresford, SD (43.048568°, −96.894815°); Tracy, MN (44.347556°, −95.543803°); Waseca, MN (44.060210°, −93.546401°).

The climate pattern from west to east shows increased rainfall. The annual average for the past 10 years is 704 mm for Beresford, 711 mm for Tracy, and 940 mm for Waseca.

## 3. Materials and Methods

### 3.1. Sampling

Isotope samples were collected monthly from three locations March 2016 to December 2017: Beresford, SD in the Vermilion River Watershed, Tracy, MN in the Cottonwood River Watershed, and Waseca, MN in the Le Sueur River Watershed.

All three locations were associated with a research farm. Each research farm had rain gauges installed to track the exact amount of rainfall the site receives. ISCO samplers were also set up for subsurface water sample collections. At each site, isotope samples were gathered from subsurface tile drains, groundwater wells, and river sites each month during the open water season. For the Tracy site, additional samples were collected from a suction cup lysimeter, a shallow monitoring well, and a surface wetland. Precipitation samples were collected when available. Samples were stored in tightly capped bottles at room temperature prior to analysis to prevent evaporation- and condensation-induced fractionation.

Figure 2 shows the sampling locations at each site. Table 1 shows the depth of each sampling location. Table 2 shows the number of samples collected each year and each

location. A bailer was used to sample the wells. The wells were bailed ten times before sampling to purge the casing and allow water to flow in from the adjacent soil.

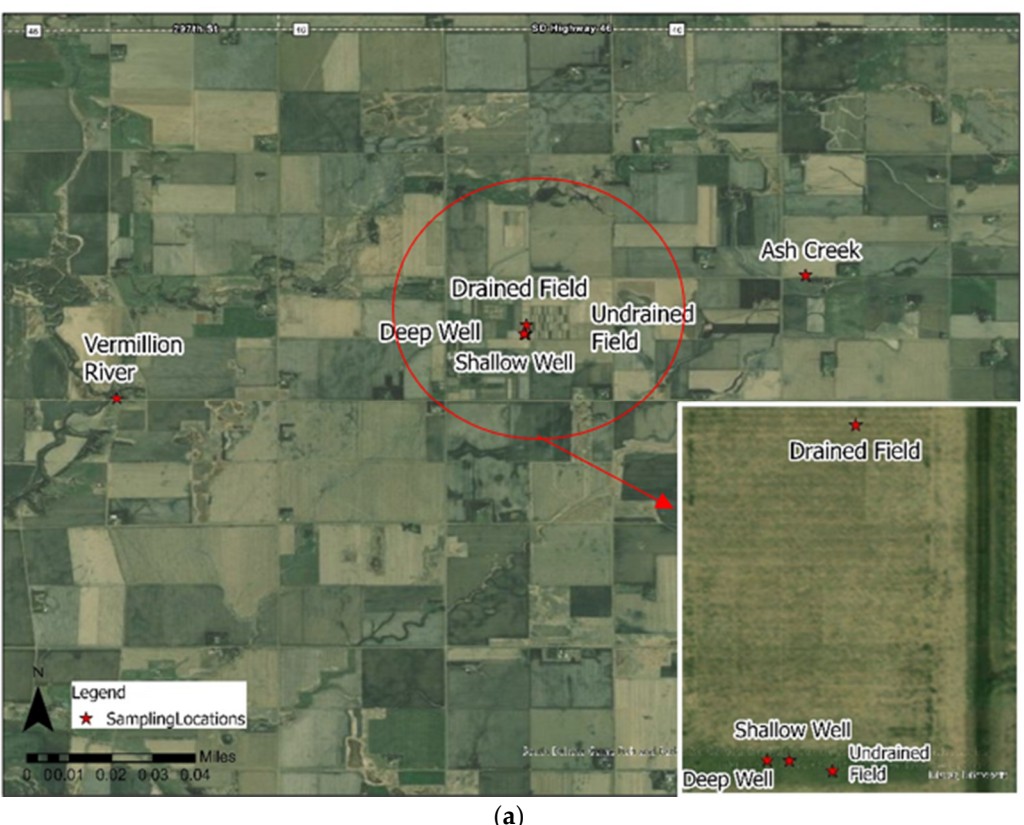

(**a**)

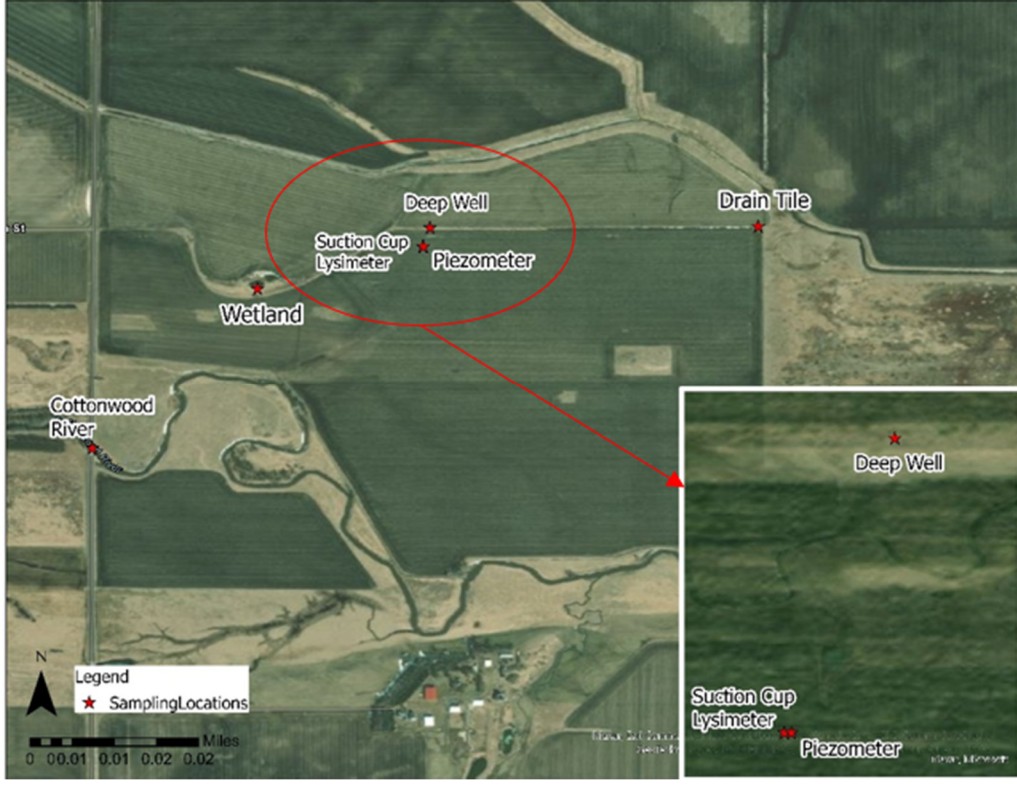

(**b**)

**Figure 2.** *Cont.*

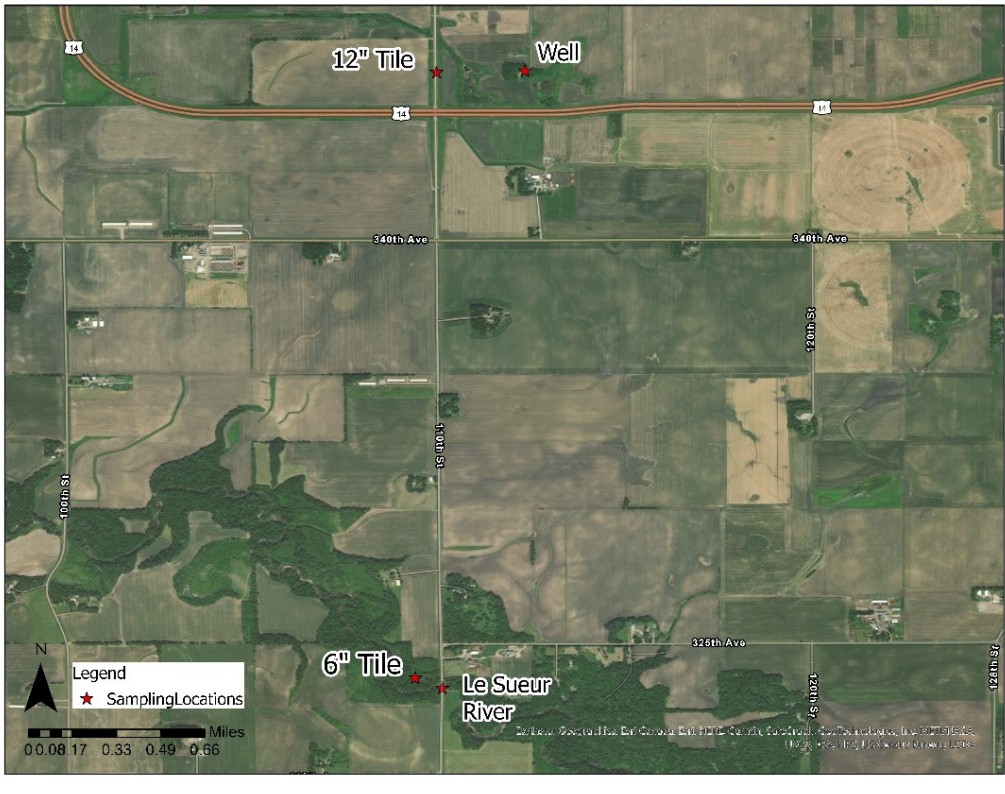

**(c)**

**Figure 2.** Sampling locations at each site. (**a**) Beresford, SD site sampling locations. Locations within the research farm is shown in the inset map on a smaller scale; (**b**) Tracy, MN site sampling locations. Locations within the research farm is shown in the inset map on a smaller scale; (**c**) Waseca, MN site sampling locations.

**Table 1.** Subsurface sampling depth at each site (from shallow to deep).

| Sampling Location | Waseca | Tracy | Beresford |
|---|---|---|---|
| Lysimeter | - | 0.76 m | - |
| Tile drains | 1.2 m | 1.2 m | 1.2 m |
| Shallow well | - | - | 1.5 m |
| Piezometer | - | 2.4 m | - |
| Deep well | 13.1 m | 10.7 m | 10.7 m |

**Table 2.** Number of samples collected in 2016/2017 at each site for each location.

| Sampling Location | Waseca (2016/2017) | Tracy (2016/2017) | Beresford (2016/2017) |
|---|---|---|---|
| Lysimeter | - | 5/2 | - |
| Tile drains | 7/9 | 7/1 | 6/7 |
| Shallow well | - | - | 4/5 |
| Piezometer | - | 6/6 | - |
| Shallow groundwater | 7/11 | | |
| Deep well | 7/11 | 7/7 | 6/6 |
| Wetland | | 7/5 | |
| Streamflow | 6/11 | 7/6 | 9/6 |

The university-level lab used a laser spectroscopy system (Liquid Water Analyzer, DLT-100, Los Gatos Research, Inc.) coupled to a PAL autosampler for simultaneous measurements of D/H and $^{18}O/^{16}O$. The precision of $\delta D$ is $\pm 1.0$‰ and that of $\delta^{18}O$ is $\pm 0.25$‰. A standard was run after every two unknown samples to correct for any instrumental drift and errors [30].

### 3.2. End Member Mixing Analysis (EMMA)

The common two end members are event water and pre-event water [31]. Stream water is commonly composed of groundwater, rainfall, overland runoff, shallow groundwater seepage, and, specifically in intensively managed agricultural landscapes, tile drainage. All the above components can be considered as event water. Assumptions to neglect certain sources is necessary in calculations to reduce the complexity of the model. Major components, such as groundwater and/or tile drainage, are commonly used.

Hydrograph separation is commonly performed with environmental tracers, conservative tracers, and semi-conservative and reactive tracers [4,6–8]. Conservative tracers are tracers that do not get affected by biogeochemical processes during transport, such as chloride and stable isotopes. Non-conservative tracers, such as nitrate, change concentration during the transport.

Mass balance/hydrograph separation with isotopes date back to late 1960s [31–36]. Five assumptions were refined:

*"(1) The isotopic content of the [precipitation] event and the pre-event water are significantly different; (2) The event water maintains a constant isotopic signature in space and time, or any variations can be accounted for; (3) The isotopic signature of the pre-event water is constant in space and time, or any variations can be accounted for; (4) Contributions from the vadose zone must be similar to that of groundwater; (5) Surface storage contributes minimally to the streamflow."* [31]

A mass balance hydrograph separation model can be a two-component model or a three/multiple-component model [37]. The two-component model separates the pre-event component and the rainfall input component. Two-component models use the idea of mass balance in standard mixing conditions and equal the amount of a certain stable isotope (concentration ∗ volume) before and after the event, thus partitioning the stream hydrograph into pre-event and rainfall components. For three- or multi-component hydrograph separation, an additional tracer is needed, or a measurement of one-flow component is required [31]. One of the challenges is the accurate identification of endmembers, as this will influence the calculated event/pre-event water fractions [31]. Groundwater, vadose zone, precipitation, and stormwater may all have a contribution to the flow samples, among which snow and rain on snow will further complicate the condition. Using oxygen and hydrogen isotopes is ideal for hydrograph separation models because they are naturally added in precipitation events and are only going to change due to mixing once free from evaporative exposure [37].

The expression provided in Isotope Tracers in Catchment Hydrology [37] was given by Pinder and Jones [38] and Dincer [34]. It was adopted by Sklash [39] and Kennedy [40], and expressed in Equation (3):

$$\delta D_E V_E = \delta D_{PW} V_{PW} + \delta D_R V_R \tag{3}$$

where D represents deuterium, V represents volume, subscript E represents the total runoff due to the precipitation event, PW represents the pre-event, and R represents the newly added water. In Kendall & McDonnell [37], the above equation is combined with the mass balance among the three components: total runoff equals the sum of the other two; therefore, Equation (3) becomes Equation (4):

$$V_{PW} = \left[ \frac{\delta D_E - \delta D_R}{\delta D_{PW} - \delta D_R} \right] V_E \tag{4}$$

### 3.3. Mean Transit Time (MTT)

In addition to hydrograph separation, stable oxygen and hydrogen isotopes have also been used to gain a better understanding of the MTT of water [17,18].

A lumped parameter method was used for the MTT analysis [20]. Convolution integral with lumped parameters mathematically describes the transport of conservative tracers through a watershed.

The convolution integral is expressed in Equation (5):

$$\delta_s(t) = \int_0^\infty TTD(\tau)\delta_P(t-\tau)d\tau \tag{5}$$

where $\tau$ is the lag time between precipitation and streamflow isotopic composition, and therefore, $(t-\tau)$ is the time of entry into the system, $\delta_s(t)$ is the lagged response of isotopic signal of receiving water (streamflow), $\delta_P(t-\tau)$ is the isotopic signal of precipitation, and $TTD(\tau)$ is the distribution of the water travel time.

Travel time distribution takes various shapes depending on the flow path and mixing mechanisms. These include, but are not limited to, piston flow, exponential flow, dispersion flow [17,24,25,41–45].

There are two main components in the equation: input function and travel time distribution. Tile drain systems promote infiltration and reduced surface runoff; the surface runoff contribution to streamflow is relatively small compared to subsurface water. It is not appropriate to directly use precipitation $^{18}$O as the input to Equation (6). Therefore, an input function, Equation (7), was necessary to adjust the precipitation $^{18}$O to a more appropriate recharge water isotopic signature [46,47].

$$\delta_s(t_i) = \frac{[N\alpha_i P_i(\delta_i - \delta_{GW})]}{\sum_{i=1}^N (\alpha_i P_i)} + \delta_{GW} \tag{6}$$

$$\alpha = \left[\sum_w (P_i\delta_i) - \delta_{GW}\sum_w (P_i)\right] \Big/ \left[\delta_{GW}\sum_s (P_i) - \sum_s (P_i\delta_i)\right] \tag{7}$$

where $\alpha$ is the infiltration coefficient, $i$ is the $i$th month, $N$ is number of years for which precipitation was collected, $P$ is total precipitation for the corresponding month, and $w$ and $s$ represent winter and summer, respectively.

This function assumes that recharged water has the same isotopic composition as model input water. Input function calculation is based on an infiltration coefficient, mean groundwater (assumed to be recharge water) $^{18}$O composition, and weighted summer/winter precipitation $^{18}$O. The infiltration coefficient was calculated using average tile drain flow $^{18}$O signature. In this paper, summer months were assumed to be April to September and winter months were assumed to be October to March.

More to the precipitation $^{18}$O, on the days where precipitation does not occur, or precipitation sample is not available, the isotopic signature can be calculated using Equation (8) [48]:

$$\delta^{18}O = (0.521 \pm 0.014)*T(C) - (14.96 \pm 0.21) \tag{8}$$

where $T(C)$ is the average temperature of the day.

This equation represents a linear relationship between temperature and $O^{18}$ abundance. Local linear relationships can be developed with local temperature and precipitation isotopic data to have a more accurate estimate.

For the travel time distribution, the literature suggests that an exponential-piston-flow model is suited for describing water transit time [17,20]. The exponential-piston-flow model contains both piston flow and exponential flow features in the same catchment, where the piston-flow model assumes that the catchment has no hydrodynamic dispersion or mixing and has high linear flow velocity and the exponential model describes the flow pathway through a homogeneous and unconfined aquifer with constant thickness receiving uniform recharge. The model distribution is described by Equation (9) [17,19,20]:

$$\begin{aligned} TTD(\tau) &= \tfrac{\eta}{\tau m}\exp\!\left(-\tfrac{\eta\tau}{\tau m}+\eta-1\right) && for\ \tau \geq \tau m\left(1-\eta^{-1}\right) \\ TTD(\tau) &= 0 && for\ \tau \leq \tau m\left(1-\eta^{-1}\right) \end{aligned} \tag{9}$$

where $\tau m$ is the mean transit time and $\eta$ is calculated as the total volume/exponential volume, also known as the EPM ratio plus 1. The EPM ratio is the ratio of the length of the area at a water table not receiving recharge to the length of area receiving recharge.

For comparison, a sine-wave model was also selected for this study. When plotted against time, isotopic signatures of both the annual precipitation and surface water appeared to take the shape of a full sine curve. Assuming the waters represent a steady-state, well-mixed reservoir where the amplitude value was extracted from both curves and used in Equation (10):

$$T = \omega^{-1} \left[ \left( \frac{A}{B} \right)^2 - 1 \right]^{\frac{1}{2}} \tag{10}$$

where T is the residence time (days), $\omega$ is the angular frequency of variation ($2\pi/365$ days), A is the input (precipitation) amplitude, and B is the output (surface water) amplitude.

However, there were sampling limitations in Minnesota due to the weather. Under frozen conditions, some tile drains and subsurface lysimeters were not available for sampling. Thus, this impacted the estimation of the amplitude of the output sine curves. This would add errors to the MTT estimation.

*3.4. Sensitivity Analysis*

Due to data availability, limitations and uncertainties were associated with the hydrograph separation analysis. The samples were at fixed locations collected monthly. However, this was not representative of the entire watershed temporally and spatially. Each tile drain in the watershed receives water that moved through different landscapes and soil profiles. Varying evaporation and condensation conditions result in different isotopic signatures for tile drain waters. To investigate the reliability of the model, a sensitivity analysis was performed.

For each site, each input parameter (source water $\delta D$ and $\delta^{18}O$) was increased or decreased individually where other inputs were held the same. The corresponding output of tile drain water fraction in percentage was plotted against percent change in input parameters. The curves visually represent the magnitude of impact of each input parameter.

**4. Results**

Average annual rainfall during the study period for Beresford was 790 mm, for Tracy was 870 mm, and for Waseca was 1150 mm. Comparing precipitation isotopic signatures helps to understand if the source of water was from the south (typically Gulf of Mexico air) or north (polar cold air). This information can help explain the isotopic signatures of the other sampling locations given that precipitation is the driving source water.

A scatter plot was created for precipitation data from all three sites (Figure 3a) for direct visual comparison. Tracy and Waseca data were plotted within the same range and Beresford data were plotted further into the first quadrant. Boxplots were also created for both $\delta D$ and $\delta^{18}O$ (Figure 3b,c) of the precipitation samples. Both boxplots show that Tracy and Waseca had similar $\delta D$ and $\delta^{18}O$ distribution, meaning that the precipitation clouds had similar evaporative signatures, possibly from the same source. Precipitation clouds at Beresford showed the widest range and the heaviest isotopic signatures (perhaps more of a gulf stream effect during summer months). This is explained by the rainout effect that as clouds move up in latitude, precipitation condensation removes heavier isotopes and leaves the clouds isotopically lighter [49,50].

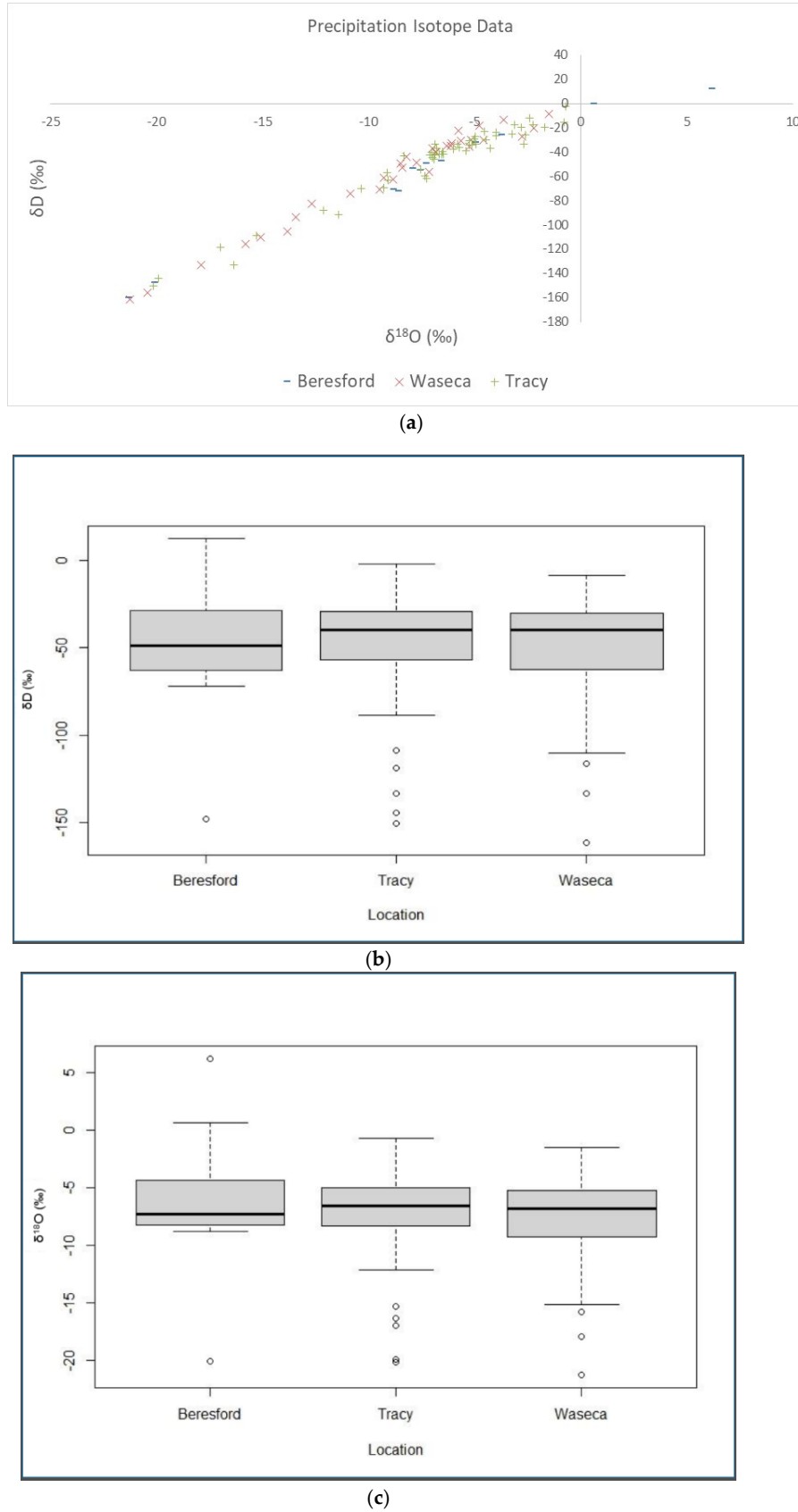

**Figure 3.** Precipitation isotopic signature plots. (**a**) Isotopic signature plots in 2016–2017 from Beresford, Tracy, and Waseca; (**b**) Boxplot for Delta D from three site using 2016 and 2017 daily precipitation data; (**c**) Boxplot for Delta O18 from three site using 2016 and 2017 daily precipitation data.

### 4.1. Meteoric Water Line

Local meteoric water lines (LMWL) were established for all three sites (Figure 4). The meteoric is expressed in Equation (11) and Table 3 summarized the intercept and slope of each site.

$$y = ax + b \qquad (11)$$

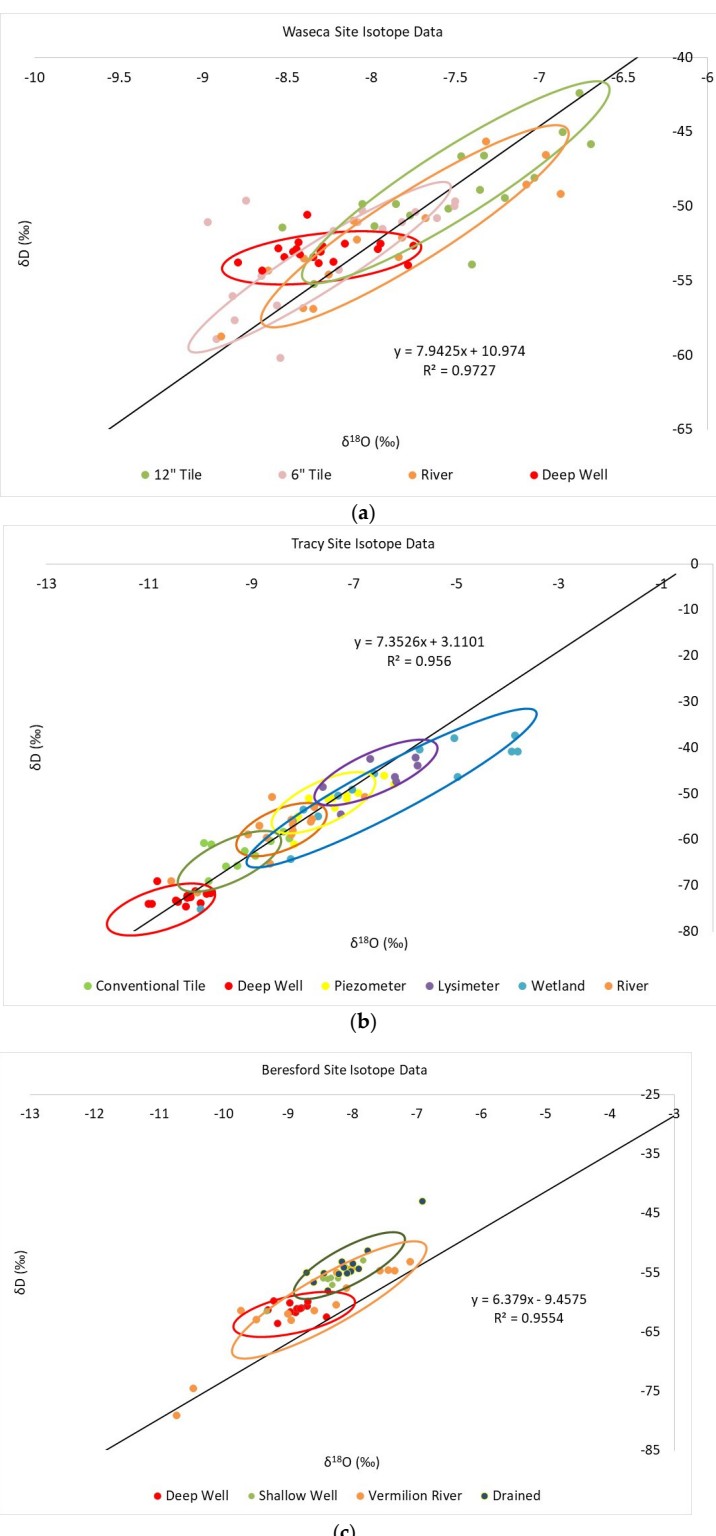

**Figure 4.** LMWL and water sample isotope plots for each site. (**a**) Waseca site; (**b**) Tracy site; (**c**) Beresford site.

**Table 3.** Parameters of meteoric water lines.

| Scheme | Slope (a) | Intercept (b) |
| --- | --- | --- |
| Global MWL | 8 | 10 |
| Waseca LMWL | 7.94 | 10.97 |
| Tracy LMWL | 7.35 | 3.11 |
| Beresford LMWL | 6.38 | −9.46 |

Among the three sites, the Tracy site data deviated the most. The differences between sampling locations are very well demonstrated on the Figure 4b. Wetland and deep well data lie on the opposite ends of the plot. They show the most difference in isotopic composition. The wetland water showed more of an evaporative signature than the rest of the locations, which is typical of surface water evaporation. The wetland water isotopic signature ranged from deep well to evaporative water. It is likely that the wetland received groundwater input to sustain the long-term standing water table. The deep well water had the least evaporative signature.

Waseca and Beresford site data points plotted more tightly than Tracy. Field investigation suggested that at Waseca, the 6″ tile drain conveyed shallow groundwater. Therefore, it is not surprising that the isotopic signatures from the 6″ tile drain were similar to the deep well. River and tile drain data both showed evidence of evaporation with the tile drain water showing a little more evaporative influence. The deep well and the 6″ tile drain isotopic signatures intercept with all the other locations. A groundwater isotope signature was present at all other locations. Compared to the Tracy site, groundwater appears to play a larger role on the watershed hydrologic characteristics.

Beresford had almost all the water samples plot above the LMWL. This is not commonly seen, but not impossible. The literature suggests that this phenomenon happens in low humidity regions when re-evaporation of precipitation increases water vapor masses with an even lighter isotopic composition [51]. Groundwater also appeared to have a significant influence upon the river water signature of this watershed.

*4.2. EMMA*

The two-component hydrograph separation model was applied to all three sites. The number of endmembers that can be solved is limited by the number of variables measured due to mathematical limitations. The results are shown in Figure 5. Uncertainty existed in the two-component model, which was quantified using a sensitivity analysis. River sampling locations were chosen to be far away from the tile drain outfall and groundwater seepage points to minimize the impact of any partial mix. The analysis assumed that the streamflow was a well-mixed system.

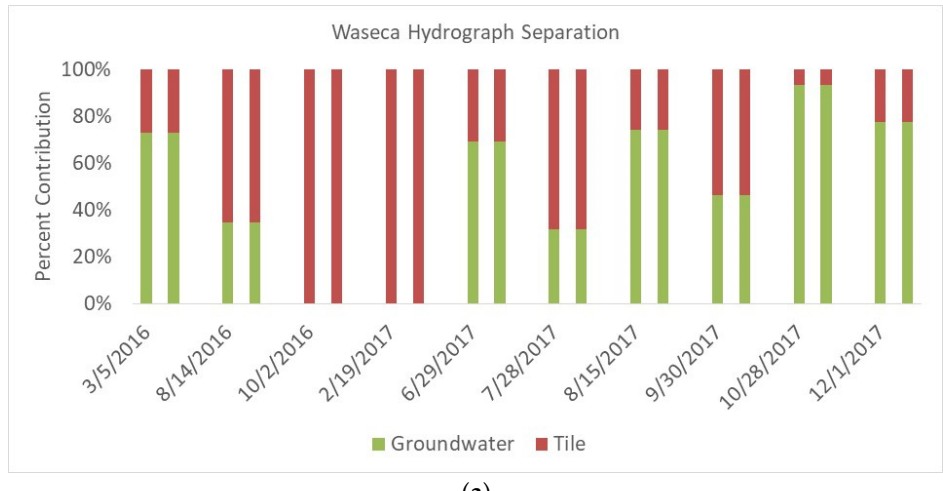

(**a**)

**Figure 5.** *Cont*.

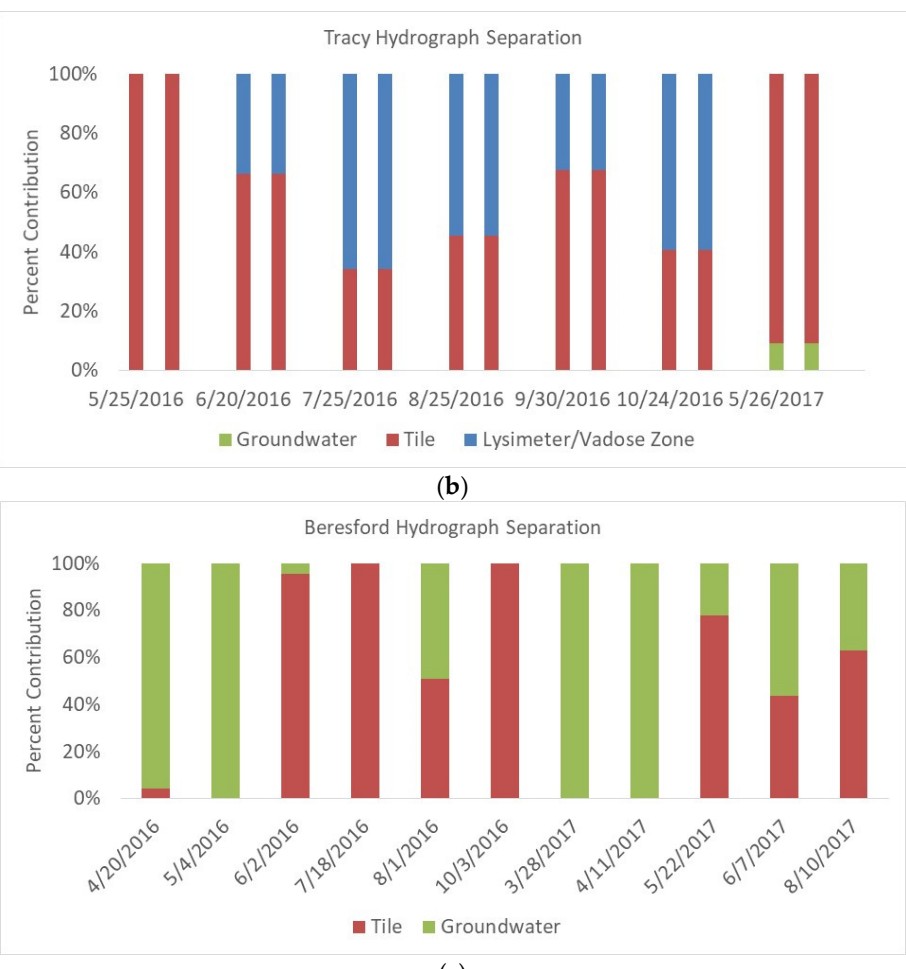

**Figure 5.** Two-component hydrograph separation for the three sites. GW stands for groundwater. The y-axis is the ratio of component assuming total volume is one. (**a**) Waseca site; (**b**) Tracy site; (**c**) Beresford site.

Based on field observation, on days with no precipitation, tile drain flow and groundwater seepage were the major inputs to streamflow. Other minor inputs include surface runoff and vadose zone water from a long-term scale. The tile drain flow isotopic signature was always used as one of the endmember components when available at all three sites given the goal of characterizing tile drain flow contributions to streamflow. The dates were selected when streamflow, tile drain, and deep well (or wetland at Tracy site) were all available. No precipitation and/or major surface runoff was observed on the selected days.

At the Tracy site, lysimeter isotopic signatures were used to represent the vadose zone component based on the interpretation of LMWL in Figure 5b. The first attempt did not include vadose zone soil water and we attempted to partition out groundwater and tile drain flow. However, on most of the days, the river water had an isotopic signature lighter than both phreatic zone sources (Figure 5b), where vadose zone soil water became one of the components. On days when groundwater had a more significant impact on streamflow, this was around 9%. However, tile drain water was responsible for more than 60% of the streamflow on days data were collected. This should not be interpreted as an annual average, as tile drain flow rate varied from day to day including some days with no flow.

At the Waseca and Beresford sites, no water samples representing surface runoff or vadose zone soil water were taken. Therefore, only groundwater and tile drain flows were used as the mixing components. Both locations agreed with the LMWL plot, and groundwater showed a bigger influence on streamflow than Tracy. Waseca and Beresford groundwater showed an average of 50% and 51% contribution to streamflow on days data

were collected. Correspondingly, Waseca and Beresford tile drain flow influenced 50% of streamflow in the LeSueur River and 49% for Vermillion River. Overall, at all three sites, the streamflow isotopic signature was clearly influenced by tile drain flow.

The tile drain contributions agreed with Rozemeijer and others [5] (80% to 28%) and Tomer and others [6] (68.6%) on the tile drain flow contribution to the total outflow.

### 4.3. Mean Transit Time

Linear relationships (Equation (3)) between temperature and $\delta^{18}O$ were established for all three sites (Figure 6). This relationship was used to determine the sine-wave function fit for the precipitation data for the model input function.

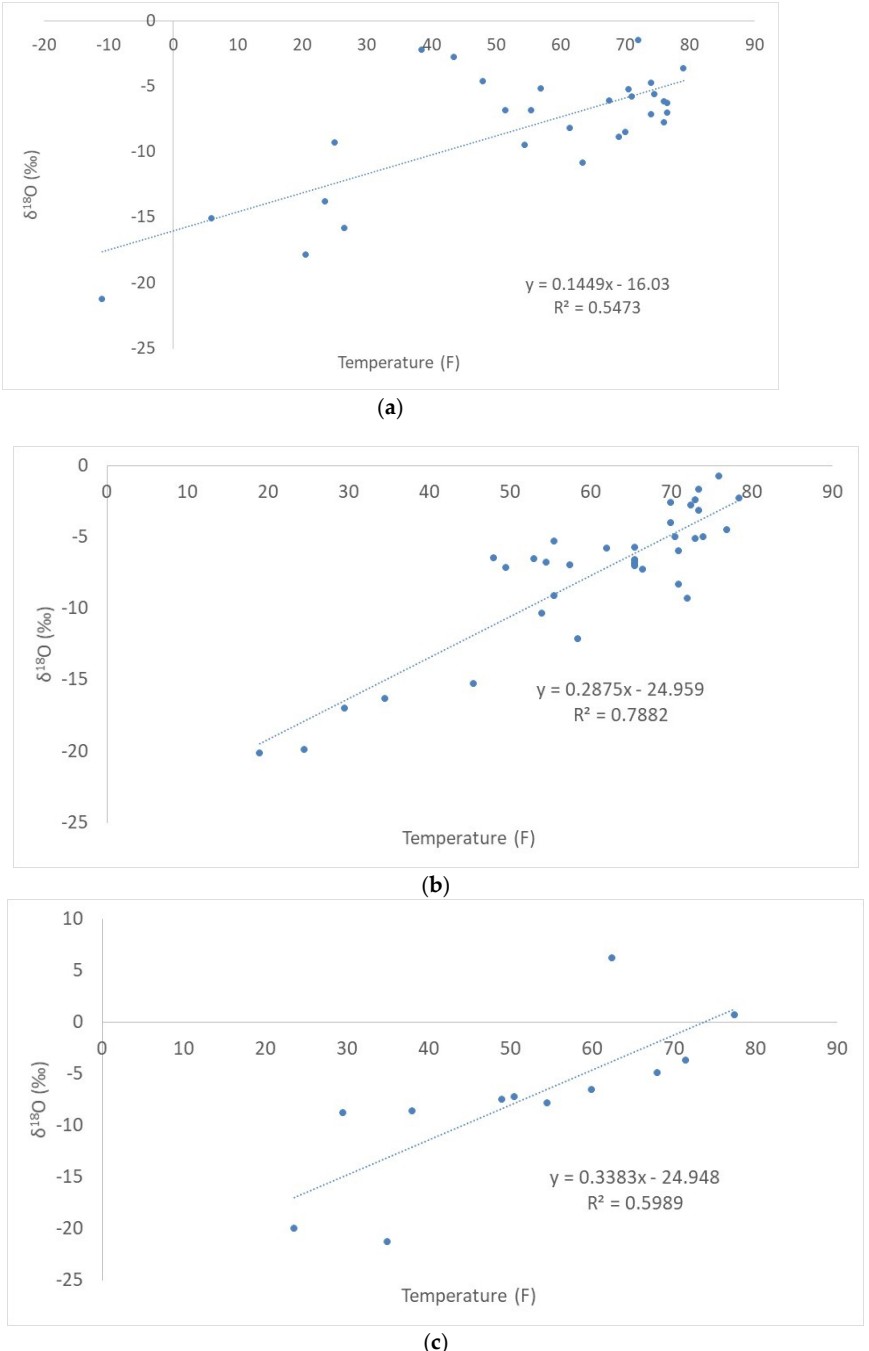

**Figure 6.** Linear relationship between temperature and $\delta^{18}O$ at all three sites. (**a**) Waseca; (**b**) Tracy; (**c**) Beresford.

Input function was obtained by calculating weighted monthly $\delta^{18}$O based on daily weather monitoring data using the above equations in the figures, and then adjusted with summer and winter infiltration rates (Equation (9)). Equation (8) explains the assumption of the input water being mostly groundwater. Since tile drain flow contributed a large percent of the streamflow at these three sites (Figure 5) and vadose zone soil water was not sampled and, thus, not well-represented, recharge water (symbol as $\delta_{GW}$) was assumed to be tile drain water with an averaged isotopic signature. The calculated infiltration coefficient for Waseca, Tracy, and Beresford are, respectively, 0.39, 0.54, and 0.16 using tile drain source water as recharge water to represent $\delta_{GW}$. Since the infiltration coefficient is the ratio of summer and winter (soil thaw and melt) infiltration amount, the number means winter infiltration exceeds summer infiltration by $1/\alpha$ times. Taking Waseca groundwater recharge water as an example, the infiltration coefficient of 0.25 means winter infiltration to groundwater exceeds summer infiltration by 4 times. This makes sense, as in winter, snow melt infiltrates into the soil and in the summer, a larger portion of soil water is lost due to evapotranspiration.

The shape of the weighted monthly $\delta^{18}$O curve suggests an absolute sine-wave function as shown in Equation (12). The approximate function parameters for each site were calculated using the least-squares method. The parameters are shown in Table 4.

$$\delta^{18}O = A|\cos(\omega t - \varphi)| + S \tag{12}$$

where the radial frequency constant, $\omega$, in the sine-wave function is $2\pi/12$ radians d$^{-1}$.

**Table 4.** Input function sine wave parameters and the root mean square errors (RMSE).

|  | *A* | ]$\varphi$ | *S* | **RMSE** |
|---|---|---|---|---|
| Waseca | 4.5 | 3.8 | −8.5 | 0.36 |
| Tracy | 7 | 3.35 | −8.5 | 0.57 |
| Beresford | 10 | 4.56 | −6 | 1.89 |

The MTT was solved using Excel Solver least-squares method. The results are presented in Table 5. The Exponential-Piston-Flow model parameter $\eta$ is the ratio of the total flow volume to the volume of the exponential flow. When $\eta$ approaches 1, the system is close to 100% exponential flow; when $\eta$ is larger (greater than 6), the system is closer to piston flow.

**Table 5.** Water mean transit time at each site.

| **Location** | **EPM MTT (Month)** | **EPM η** | **EPM RMSE** | **Sine Wave (Month)** |
|---|---|---|---|---|
| | | Waseca | | |
| 12″ Tile | 3.01 | 1.01 | 0.36 | 3.85 |
| 6″ Tile | 4.24 | 1.07 | 0.34 | 6.22 |
| Well | 4.76 | 1.08 | 0.35 | 16.36 |
| River | 3.76 | 1.06 | 0.29 | 2.74 |
| | | Tracy | | |
| Tile | 6.01 | 1.41 | 0.77 | 17.6 |
| Well | 15.75 | 1.81 | 0.28 | 21.81 |
| Lysimeter | 2.98 | 1.10 | 0.50 | 14.38 |
| Piezometer | 4.77 | 1.08 | 0.60 | 13.51 |
| River | 4.67 | 1.27 | 0.40 | 7.34 |
| Wetland | 4.47 | 1.04 | 0.28 | 4.66 |
| | | Beresford | | |
| Tile | 5.3 | 1.14 | 0.39 | 21.21 |
| Shallow well | 6.94 | 1.23 | 0.25 | 25.47 |
| Deep well | 9.78 | 1.37 | 0.23 | 41.93 |
| Vermillion River | 4.27 | 1.11 | 0.50 | 10.37 |

### 4.4. Sensitivity Analysis

Evaluating the robustness of the system improved the understanding of the uncertainty of the analysis. Sensitivity analysis was applied on all three sites and presented visually in graphs below (Figure 7). Each variable changed was listed in the legend of each graph, where O represents $\delta^{18}$O and D represents $\delta$D. All three graphs indicated the same pattern, where changes in $\delta^{18}$O had little impact on the estimated tile drain flow proportion. A small change in $\delta$D resulted in a big shift in the estimated tile drain flow contribution to the streamflow. The confidence of the model is limited by the accuracy of $\delta$D sampling and analysis.

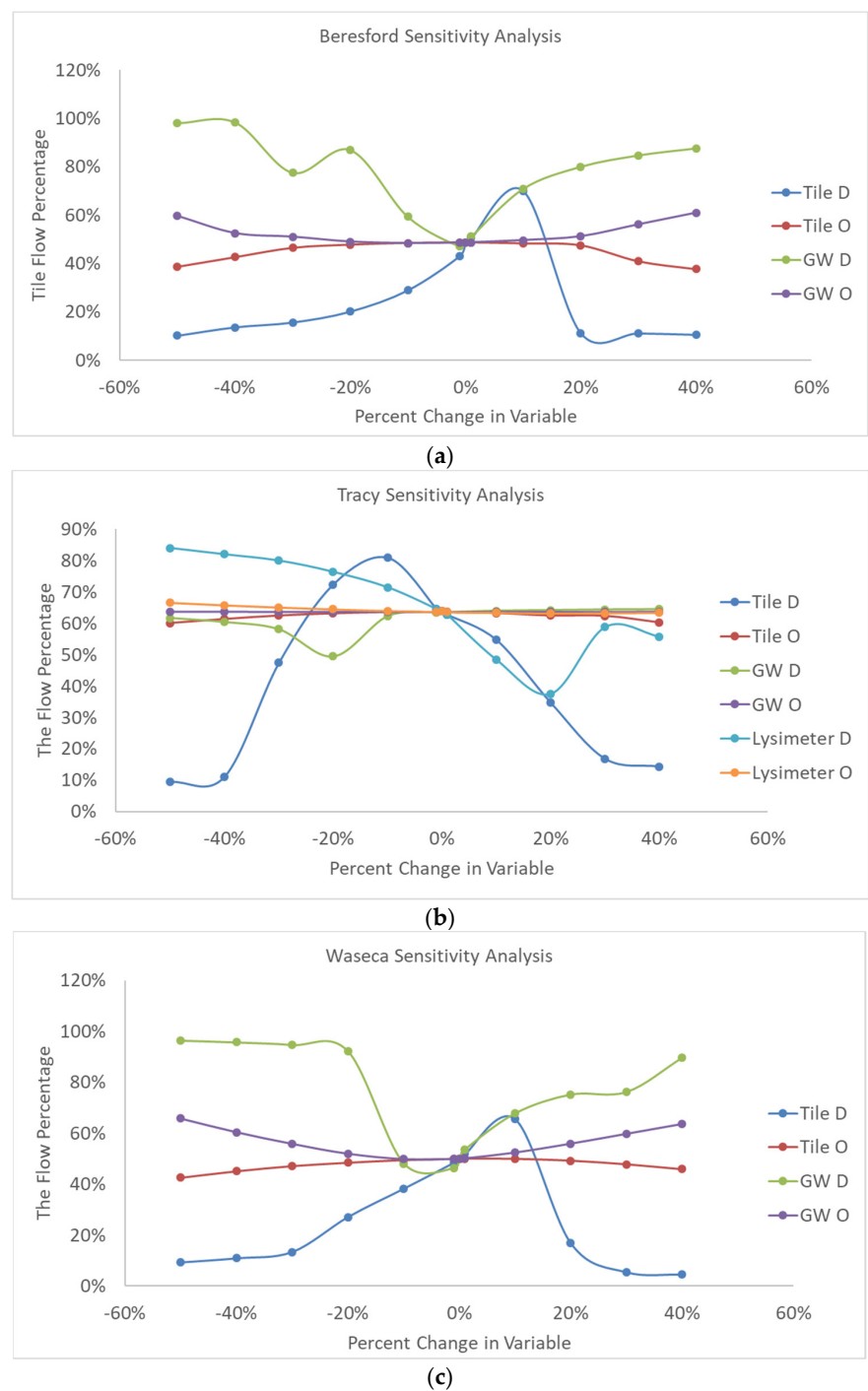

**Figure 7.** Sensitivity analysis where each variable was changed and the change in the estimated tile flow percentage was compared. (**a**) Waseca site; (**b**) Tracy site; (**c**) Beresford site.

## 5. Discussion

One of the major factors that limited this analysis was the number of variables. As discussed in the previous section, the number of components was governed by the number of variables measured. Two tracers ($\delta D$ and $\delta^{18}O$) were used; however, considering the complexity of the system, where streamflow receives major input flow from shallow and deep groundwater, tile drain flow, overland runoff, and direct precipitation, two end members did not fully represent the other components. Therefore, in Waseca, there were days where groundwater input appeared to not exist. Mathematically speaking, if streamflow isotopic signatures are a result of the mixing, the streamflow isotopic number should be in between the two sources' isotopic numbers. However, if the two sources are both greater or less than the streamflow isotopic number, this causes one source to dominate. Another source contributed to the streamflow to balance out the mixing, which was not represented in the two-component model.

Common belief is that water enters the drain tiles from the top by gravity; however, this is a misunderstanding. Nevertheless, in Figure 6, the relative plot locations between groundwater wells and tile drains further explained how water enters the tile drains. At all three sites, there was a certain level of overlapping between well water isotopic signatures and tile drain water isotopic signatures, meaning that tile drain water has groundwater intrinsically mixed into the tile drain as the water table rises. The zone between vadose, tile drain, and shallow groundwater is dynamic and can be influenced by the residence of existing water and new precipitation additions [42]. Because water enters the tile drain from the bottom due to the water table rising, this water can have a mix of existing groundwater, vadose water, and new precipitation, which creates a new composite mixture.

The water MTT calculations agreed with the LMWL plot (Figure 6). Both Waseca and Beresford plots indicated a large groundwater impact, which was also shown by the MTT estimation that other water MTTs were closer to the well water MTT compared to the Tracy site. However, the sine wave model predicted a different MTT compared to the lumped parameter model. The sine wave model is highly dependent on the accuracy of the annual amplitude estimation. Due to the sampling limitations, the amplitudes could not be accurately estimated with the available data. Therefore, it is likely that at locations where more data were available, the two MTTs were more comparable.

The MTT of tile drain water indicates the average time that precipitation infiltrated into the soil and raised the water table, and then moved into the tile drain. This is the time given to the microbial community for nutrient transformation and/or removal before the excess nutrient is transported to the stream. This movement happens in the vadose zone. The unsaturated condition limits the process of denitrification. This likely explains why relatively high concentrations of nitrate-nitrogen can be found in Midwestern streams, where tile drains are used to enhance crop production.

## 6. Conclusions

From west to east, Beresford, Tracy, and Waseca have increasing average annual precipitation and relative humidity. This climate gradient can be seen in the isotopic signature distribution on the LMWL, where the signatures were a lot closer to each in less humid regions. However, this climate gradient does not seem to have a huge impact on the streamflow sources. The tile drain contribution remained large, and this agrees with other documented studies. For the MTT estimation, where there is an adequate amount of data, the lumped parameter analysis and the sine wave methods were comparable. However, the sine wave methods for less humid regions showed difficulty due the lack of variation in isotopic signatures; for example, at the Beresford site, the small range of the isotopic signatures made sine wave estimation difficulty, driving up the RMSE of the estimation, making the results less accurate than the lumped parameter analysis. Vadose zone water transport was complicated. This study was also limited by the amount of data available and the duration of the study. Tile drain sampling highly depends on the weather. The tile drains did not flow continuously to allow sampling at the desired frequency.

Future research with a potentially longer duration is recommended to further analyze the characteristics of tile drain water and characterize water from different soil horizons to relate soil evaporation to the transport behavior.

**Author Contributions:** Conceptualization, Funding acquisition, Writing—review & editing: J.S.; Formal analysis, Investigation, Writing—original draft: L.Z.; Conceptualization, Supervision, Writing—review & editing: J.M. All authors have read and agreed to the published version of the manuscript.

**Funding:** This research was funded by the Minnesota Corn Growers Association.

**Institutional Review Board Statement:** Not applicable.

**Informed Consent Statement:** Not applicable.

**Data Availability Statement:** Lu Zhang's dissertation is published online at the following link: {Hydrologic Impacts of Tile-Drained Landscape and Isotope Tracer Analysis} (umn.edu).

**Conflicts of Interest:** The authors declare no conflict of interest.

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
