# Peer review of "Exploring a Climate Gradient of Midwestern USA Agricultural Landscape Runoff Using Deuterium (δD) and Oxygen-18 (δ18O)"

_water, doi:10.3390/w14101629_

Round 1

Reviewer 1 Report

This paper reports an interesting case study based on very valuable multi-year isotopic dataset. It provide new insights into the influence of landscapes on runoff generation. Overall, it is well written. However, there's some mistakes in contents, structure and languages. My main concern is what implication can be obtained for the "climate gradient" mentioned in the title. More examples should be provided in the Introduction section to help readers better understand the backgrounds. Please consider the comments and suggestions below before submitting the revised version.

Title: superscript for "δ18O".

Abstract: the title emphasizes the "climate gradient", but no description about this concept occurs in the abstract. It's not reasonable. You should outline what can be inferred from isotopic analyses about the climate gradient.

Introduction: 

The introduction is too short and simple. The background and meaning of mean transit time estimation should be introduced as MTT is an important scientific issue discussed in this work. It is suggested to provide more relative examples on mean residence time, especially under influence of landscapes. The following papers are relative to these points and may be referred to:

  1. Rodgers et al. (2005) Using Stable Isotope Tracers to Assess Hydrological Flow Paths, Residence Times and Landscape Influences in a Nested Mesoscale Catchment. Hydrology and Earth System Sciences.
  2. McGuire et al. (2006) A Review and Evaluation of Catchment Transit Time Modeling. Journal of Hydrology.
  3. Xia et al. (2021) Revealing the impact of water conservancy projects and urbanization on hydrological cycle based on the distribution of hydrogen and oxygen isotopes in water. Environmental Science and Pollution Research.
  4. Hu et al, (2015) Seasonal recharge and mean residence times of soil and epikarst water in a small karst catchment of southwest China. Scientific Reports.

Also, literature review is suggested to be provided for hydrograph separation. You can refer to both review articles and case studies. The following ones may be considered:

  1. Klaus et al. (2013) Hydrograph separation using stable isotopes: Review and evaluation. Journal of Hydrology.
  2. Doctor et al. (2006) Quantification of karst aquifer discharge components during storm events through end-member mixing analysis using natural chemistry and stable isotopes as tracers. Hydrogeology Journal.
  3. Xia et al. (2021) Impact of human activities on urban river system and its implication for water-environment risks: an isotope-based investigation in Chengdu, China. Human and Ecological Risk Assessment.

Additionally, the properties, applications and advantages of hydrogen and oxygen isotopes are also recommended to be summarized.

Figure 1: this map looks like a screenshot and is not with a high enough resolution. Please add a large-scale sub-map (e.g., national scale) to make it clear about the location. Also the coordination is necessary. Additionally, I cannot see any information about depth in the map (the caption).

Line 69: precipitation amount is usually expressed in "mm". Please also add the source/publisher of climatic data.

Lines 73-77: move these sentences to "Introduction" section.

Figure 2: it's difficult to obtain useful information in these google map images. what do different colors mean? what are vegetation cover types? Please add compass, coordination and  proportional scale, and also add some explanations in the main text.

Line 98: what installment is used for precipitation sample collection? How many samples for different water pools are collected in total?

Line 121: the initial paper that reporting the GMWL should be the following one:

CRAIG H. Isotopic Variations In Meteoric Waters. Science, 133 (346), 1702, 1961.

Section 3.2: I don't think the basic knowledge of MWL is important in this research paper, as it is not your original method or derived from your data analysis. It is suggested to move this part to the "Introduction" or delete it. You could just state how to set up such relation in "M & M" section.

Lines 190-194: These examples can be moved to the "Introduction" section. Please not that only description of the theory/conception and applications of the methods should be included in "M & M" section.

Figure 4: In the lowest sub-plot, gamma functions of TTD are provided under different shape factors and scale factors. But it looks a little bit confusing for readers. Please note and distinguish the values of these factors of each curve.

Line 269: what was biased? Please complete the sentence.

Section 3.5: the authors point out the possible source of uncertainty. That's correct. However, what is more important is the quantification of uncertainty. I suggest to use some mathematical method such as Gaussian error propagation.

Lines 286-287: please use the unit mm for precipitation amount.

Figure 5(b,c): unit should be added for Y axis. Also, please note what percentile the boxes and whiskers represent in the box plots.

Line 349: where is the quantified uncertainty? I cannot see it in Fig. 7.

Line 397: please use the same form for "δ18O" through the whole paper.

Section 4.3: It is suggested to add the goodness of fit to the results from sine wave regressions.

Lines 452-453: what is the mean of "should be in between the two sources"? I believe there must be errors in this expression. Please correct the sentence.

Lines 480-482: grammar error exists for this sentence.

Lines 482-483: please check the grammar error. "where" instead of "were"?

Conclusion: 

This section should be concise and short. Please avoid emphasizing the advantages of methods and repeating what methods have been used. Just outline the most important findings and new insights from your data.

Reviewer 2 Report

General remarks
I.      Abstract needs restructuring: what is new, which scientific issue will be addressed.
II.    Introduction very brief. No real introduction to a scientific problem that will be solved. 
III.     What is the research question that will be addressed in the manuscript? Why should the reader read this paper?
IV.      Climatic data (for at least 10 years)  must be included in the introduction 

V.Rain sampling methodology is absent, please add in the relative section; add quality control for isotopic analyses and the reference for the lab.
VI.     The isotopic record shown is overinterpreted; only one succession with a limited number of measurements. It is s bit overdone; more data is needed to support the interpretation given.
VII.    The referencing in the introduction and discussion could (should) be extended.
VIII.   Well-drafted figures, field data well documented. More graphs needed.

Reviewer 3 Report

- The section "Conclusions" includes mainly a summary of the methods used in this study, but no specific results are described. 

Reviewer 4 Report

The manuscript titled “Exploring a climate gradient of Midwestern USA agricultural landscape runoff using Deuterium (δD) and Oxygen-18” is within the scope of the journal. Quantify the hydrological processes using stable isotope is an interesting topic to the audiences of the hydrology community. This paper is generally well written. However, some flaws limit the quality of the study:

  1. The gap between previous studies and the present work is not fully stated;
  2. The choose of the potential end-members seems arbitrary, which is the basis of this study.
  3. Why was the two end-member model employed in this study? In the introduction section, the authors argued that “tile-drain”, “deep groundwater” and “vadose zone water” might be the potential end members of streams.
  4. The representativeness of the data is another concern, which were collected from very limited locations.

some specific suggestions

  1. L20 dataset;
  2. L32 Check the reference information of “Schottler and others (2013)”;
  3. L50 delete “percent”;
  4. L47-54 adding the reference here to support the statement;
  5. L62 delete the “,” before the first “and”;
  6. The title of Fig. 1is not correct. No information on sampling depth;
  7. L73-77 move this kind of content to the introduction section. The same to L151-155 and L “190-194”;
  8. L92 “from… to…”;
  9. L102 “from the wells”
  10. The reason for the different sampling strategies used in the three sites shown in Fig. 2;
  11. Save the space of Fig.2, which only expressed a little information;
  12. The information on the number of each water sources should be provided in materials and methods section. For example, the number of precipitation samples is not stated;
  13. L145-148, Is it reasonable that groundwater was classified as event water?
  14. L166 the form of the reference is not suitable;
  15. L172-174, “a tracer” or “another more tracer”? The three components of hydrography separation at least need two tracers;
  16. L285 “Results”
  17. L286 “study period”.

Round 2

Reviewer 1 Report

The paper has been well revised. I agree it to be published in the present form.

Reviewer 2 Report

The authors followed the proposed suggestions, and now the manuscript is significantly improved and appropriate for publication. My decision is to accept it in its current form.